

# A Homogeneous Earthquake Catalogue for Turkey and Surrounding Region

Onur Tan

İstanbul University – Cerrahpaşa, Faculty of Engineering, Dept. of Geophysics, Istanbul, Turkey

*Correspondence to*: Onur Tan (onur.tan@istanbul.edu.tr)

**Abstract.** A new earthquake catalogue for Turkey and surrounding region (32° - 47° N, 20° - 52° E) is compiled for the period 1900-2017. The earthquake parameters are obtained from the Bulletin of International Seismological Centre that is fully updated in 2020. New conversion equations between moment magnitude and the other scales ($m_d$, $M_L$, $m_b$, $M_s$ and M)

are determined using in the General Orthogonal Regression method to build up a homogeneous catalogue, which is the essential data for seismic hazard studies. The 95% confidence intervals are estimated using the bootstrap method with 1000 samples. The equivalent moment magnitudes ($M_w$*) for the entire catalogue are calculated using the magnitude relations to homogenise the catalogue. The magnitude of completeness is 2.9 $M_w$* and 3.0-3.2 $M_w$* for Turkey and Greece generally. The final dataset is not declustered or truncated using a threshold magnitude because of motivation for generating a widely

usable catalogue. It contains not only $M_w$*, but also the average and median of the observed magnitudes for each event. Contrary to the limited earthquake parameters in the previous catalogues, the 45 parameters of approximately 700k events occurred in a wide area from the Balkans to the Caucasus are presented.

## 1 Introduction

The earthquake catalogues are the first output of seismological observations. National and international catalogues are

generated by several institutions around the world for understanding the seismic activity of a region. Principally, a catalogue contains the parameters such as origin time, coordinates and focal depth. Although the magnitude of an earthquake, which is a dimensionless scale of energy release, is one of the main seismological parameters, it has different scales (types) based on different seismic wave types and determining approximation (Table 1). A catalogue may not contain all magnitude scales for an event. If an earthquake catalogue is used for just showing seismicity on a map, the magnitude type may not be important

because the differences among the values of scales are not too big for visualisation. However, magnitude scale information used in energy calculation is crucial for seismic hazard studies.

There are several unknowns for magnitude calculations of institutions due to used equations, seismic network structures, man-made mistakes etc. Both amplitude and distance constants in the magnitude equations are the major items. Although




they are specific for a region because of seismic wave attenuation in the crust and mantle, the constants calculated from the Californian earthquakes (i.e. for local magnitude by Richter, 1935; Hutton and Boore, 1987) are widely used. On the other hand, individual magnitudes are calculated at each station for an event; then they are averaged. The averaged magnitude is closely related with several factors: The number of stations, the standard deviation of the average, amplification or attenuation due to the geological structure beneath the station, the radiation pattern of the seismic waves related with the

azimuthal distribution of stations. Therefore, institutions report different magnitudes for an event. Another issue picked out in this study is about moment magnitude ($M_w$) in catalogues. $M_w$ is determined using waveform modelling for events ($M_w \geq 3.5-4.0$) that have a high signal-to-noise ratio. However, a few institutes report $M_w$ for small events ($M_w < 3.0$, i.e. 25.01.1999 13:06 $M_w$=1.8 by Cyprus Geological Survey Department; 29.05.2014 01:14 $M_w$=1.8 by the Earthquake Research Center, Ataturk University). It is clear that these magnitudes are determined by using a relationship equation, but it cannot be

proved this type of man-made faults. Consequently, there are more than one magnitude values for an event with known and unknown calculation errors, and only one scale for each event must be used in the studies based on the parametric data such as hazard mitigation analyses. At this point, essential of a homogenised catalogue with a common magnitude arises. In the last two decades, the studies on unifying earthquake magnitudes and generating improved catalogue are carried out for different parts of the earth (i.e. Grünthal et al., 2009; Chang et al., 2016; Manchuel et al., 2018; Rovida et al., 2020).


**Table 1.** Symbols for different magnitude scales in the priority order of magnitude saturation.

| | |
|---|---|
| $M_w$ | Moment magnitude |
| $M_s$ | Surface wave magnitude |
| $m_b$ | Body wave magnitude |
| $M_L$ | Local (Richter) magnitude |
| $m_d$ | Duration magnitude |
| M | General magnitude (unreported type) |

This study focuses on the earthquakes that occurred in Turkey and surroundings. The region is one of the most

geodynamically active areas on the earth and deformed among Eurasian, African and Arabian plates (Fig. 1). Both the continental collision between Arabia and Eurasia and subduction of the African Plate beneath Eurasia started in the Early/Middle Miocene (11-23 Ma). The interactions of the three plates are the major driving forces for the tectonics of the region. The plate motions result in thrust faulting in Eastern Anatolia, Caucasus and Iran, normal faulting in Western Turkey and Greece, and transform faults due to escaping to west and east (see Bozkurt, 2001 for a brief synthesis). The complex

tectonic character of the region causes a high number of earthquakes with different faulting mechanism and a wide range of focal depths.


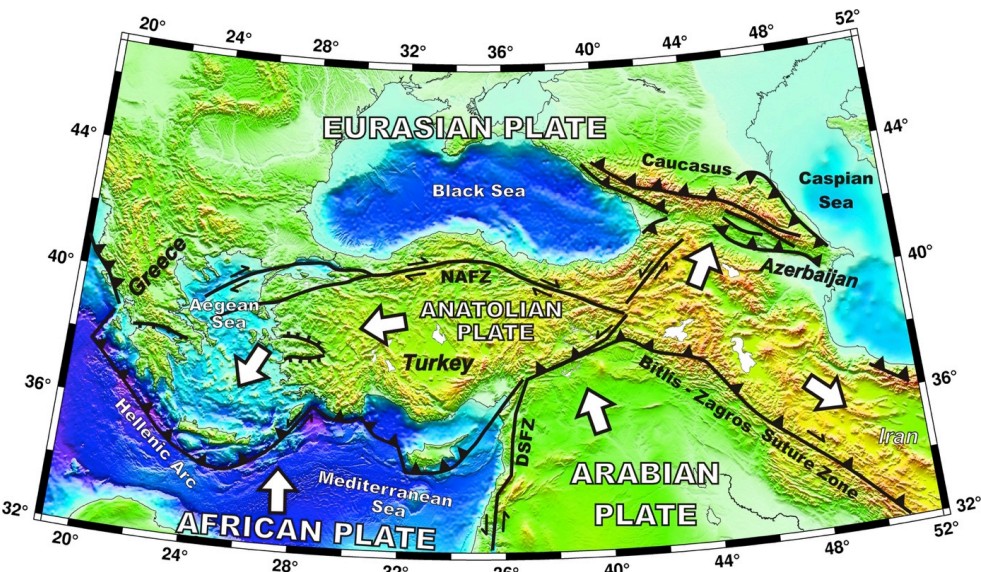

**Figure 1.** Simplified tectonic map of Turkey and surroundings. DSFZ: Dead Sea Fault Zone, NAFZ: North Anatolian Fault
Zone. Triangles indicate direction of vergence or subduction, bars indicates down-thrown side of normal faults. White
arrows indicate relative plate motions. The relief model is generated with the ETOPO1 model (Amante and Eakins, 2009).

The destructive earthquakes in Turkey and the surrounding countries along the centuries are found in the historical records.
Pınar and Lahn (1952), Ergin et al. (1967, 1971), Soysal et al. (1981), Güçlü et al. (1986), Ambraseys and Finkel (1995),
Ambraseys and Jackson (1998) compile the historical earthquakes in the region. Tan et al. (2008) present the historical
events on a digital database and publish the first catalogue that contains the focal mechanism parameters of the earthquakes
in Turkey. On the other hand, Leptokaropoulos et al. (2013) and Kadirlioğlu et al. (2018) introduce homogenised catalogues
for the Turkish earthquakes. The main component of homogenization is to obtain reliable magnitude conversion from one
scale to moment magnitude. Several empirical relations are also proposed for the region (Papazachos et al., 1997; Ambraseys
2000; Baba et al., 2000; Burton et al., 2004; Ulusay et al., 2004; Akkar et al., 2010; Deniz and Yücemen, 2010).

The motivation of this study is to build a widely usable earthquake catalogue (i.e. for geophysicists, geologists, earthquake
engineers) that contains homogenised moment magnitudes and the other seismological parameters. During the international
seismic hazard studies of the Sinop Nuclear Power Plant that is planned to construct in northernmost of Turkey, it is clearly
understood that a comprehensive homogenised earthquake catalogue for Turkey is needed for future studies. For this aim, all
earthquakes occurred in a wide area are analysed with a statistical approach, and the empirical magnitude relation equations
are obtained using a refined data set. Then, an extensive homogenised earthquake catalogue for Turkey and the surrounding



region is constructed. The distinguishing feature of the new homogenised catalogue is that it contains all earthquakes in a manageable format from Greece to Azerbaijan without removing aftershocks and truncating small events.

**2. Database and processing**

The Bulletin of the International Seismological Centre (ISC, 2020) is used as the main database to generate a new and comprehensive homogeneous earthquake catalogue for Turkey and the surrounding region. The ISC Bulletin contains a large number of parametric data for an event that occurred anywhere on the earth. Because all national and international seismological centres contribute to the bulletin, it contains not only moderate-to-large events (M≥4) but also local

earthquakes with small magnitudes (M<4). The most important feature of the bulletin is that an event with sufficient data is manually checked and relocated by a seismologist. Therefore, the latest event information in the database is two years behind in real-time (ISC, 2020). The bulletin also presents the event parameters reported by the contributor centres. The ISC finished rebuilding the entire database in 2020. The *ak135* seismic velocity model (Kennett et al., 1995) and location procedure that is recently used by the ISC is implemented to all data. Furthermore, a large number of earthquake data from

the permanent and temporary networks have been added (ISC, 2020; Storchac et al., 2017). Therefore, the latest and revised international dataset is used in this study.

The earthquake parameters in the bulletin are in the IASPEI Seismic Format (ISF, 2020). Each event has its own data block, such as origin and magnitude, contains several data types and comments. Data and comment lines have no specific flag to

identify their types, and it is not possible to read the database using a simple computer program or shell-scripts. A Fortran code is written to analyse the ISF lines using the parsing subroutines provided by the ISC. Each line in the database is checked by the different parsers to identify its data type. After determining the origin and magnitude sub-blocks of an event properly, the parameters are analysed. The overall data processing is given in the flowchart in Fig. 1. In the first step, the origin data such as time, location and focal depth are searched for the "PRIME" comment that indicates the residuals is useful to prefer the hypocentre parameters. The hypocentres determined by the ISC are always prime. If there is no "PRIME"

flag, the origin data is searched in the secondary hypocentres using a priority order for the institutes given in the flowchart. The parameters reported by the ISC are preferred first. If there is no information from the ISC, the availability of the origin parameters of the European-Mediterranean Seismological Centre (CSEM or EMSC) are tried to find (see Appendix A for the institute abbreviations). The priority of both institutes is high because they use all available data in the study area. In turn,

ISK (Kandilli Observatory and Earthquake Research Institute, KOERI) and DDA (General Directorate of Disaster Affair until September 2017; Disaster and Emergency Management Presidency - AFAD after October 2017), which are the national seismological networks in Turkey, are selected. The other institutes are used for the local events around Turkey. Besides, the earthquake information reported by ISS and GUTE is used for the pre-instrumental period (1900-1964). If the origin




parameters of an event are found in any step of this query order, this event is added to the homogenised catalogue with these

parameters.

After determining the origin parameters of an event in the selected area, the magnitude data sub-block are analysed by the

magnitude parser. The values of different magnitude scales given in Table 1 are collected. If there are two or more values for

each type, average with standard deviation and median are calculated. Selecting a magnitude value from a particular institute

is not preferred to overcome the problems such as unreported magnitude, the effect of network distribution, calculation

errors. On the other hand, we have no evidence for that an institute calculates true magnitude for an earthquake.

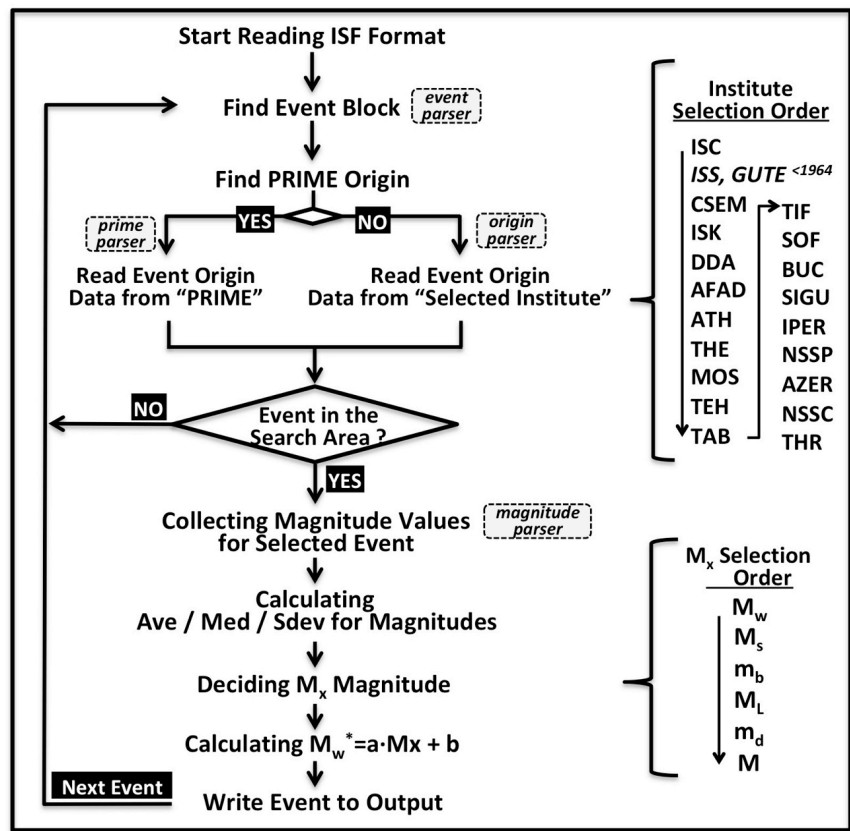

**Figure 2** Flowchart of the ISC database processing. Ave: Average (mean), Med: Median, Sdev: Standard deviation. See

Appendix A for the institute codes.



More than 7.7 million events in the ISC monthly data sets are analysed for the final catalogue. The interested area is bounded by 32°N - 47°N and 20°E – 52°E (Fig. 2). These limits cover an event that occurred 350 km away from Turkish borders to ensure the requirements of seismic hazard mitigation studies and the Turkish Regulations for nuclear power plant site studies. The study area also covers the Balkans, Black Sea, Caucasus, Syria, northern Iraq and northwest of Iran. The final catalogue contains 697,139 events occurred in the period from 1900 to the end of 2017. The instrumental period (after 1964) data is used for the statistical analyses.

The number of events (1964-2017) reported with local magnitude ($M_L$) is 443,939 (64% of the total) and it is the highest rate respect to the other magnitudes types (Fig. 3). About 35% of the events have duration magnitudes ($m_d$). Because both magnitudes types are widely determined by the national institutions, especially for local events, they are dominant in the catalogue. Body ($m_b$) and surface wave ($M_s$) magnitudes are reported for only 4.7% and 1.7% of the total events in the region, respectively. Though moment magnitude ($M_w$) is the most preferred magnitude scale for seismic hazard studies, only 0.8% of all events have $M_w$ because waveform analyses are not easy and routine process. On the other hand, the catalogue contains 28,630 (4%) events with no specified magnitude types (M). The magnitude M is mostly reported until 1990, and the number of events with M dramatically decreases after this year. Besides, approximately 2% of the annual activity is reported without a magnitude (in total 41,440 events). However, the rate runs up to 6% only in 2010 and 2011 because ~5000 events without a magnitude are reported by the TIF (Georgia) for the Caucasus earthquakes. The earthquakes with no magnitude assigned are also included in the catalogue to be useful in future studies.


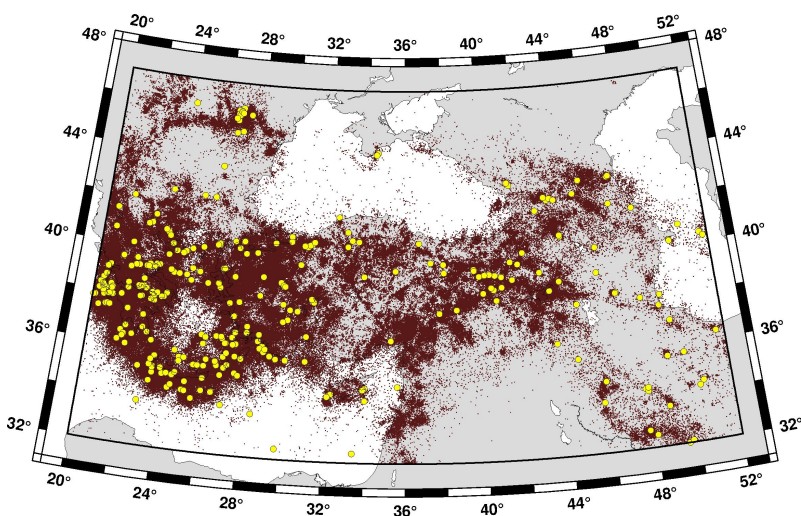

**Figure 3.** The catalogue area (solid border) and the earthquake in the catalogue occurred in the period 1900 - 2017 (dots). Yellow circles are the events with magnitude greater than 6.0.


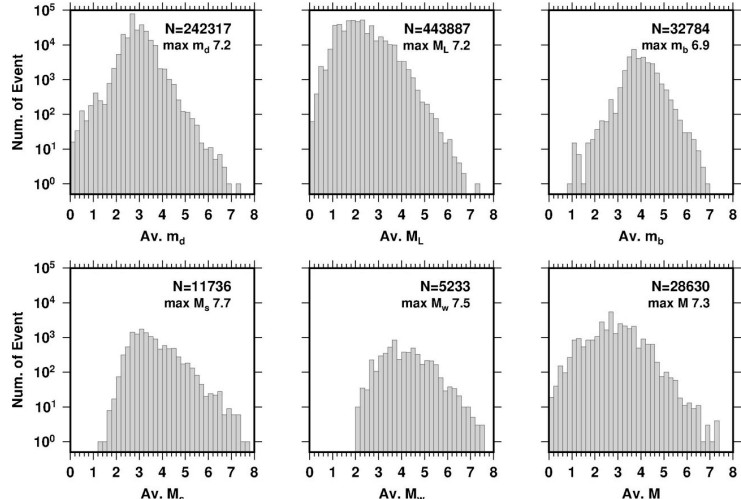

**Figure 4.** The number of events in the final catalogue for each averaged magnitudes (1964-2017). N is the total number of event for each magnitude.

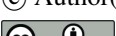



**Table 2.** Top: Number of magnitudes with zero and non-zero values between 1964 and 2017 in the catalogue. Bottom: The number of magnitude pairs with non-zero values.

|  | $m_d$ | $M_L$ | $m_b$ | $M_s$ | $M_w$ | M |
|---|---|---|---|---|---|---|
| =0 | 452,933 | 251,361 | 662,466 | 683,514 | 690,017 | 666,720 |
| ≠0 | 242,317 | 443,887 | 32,784 | 11,736 | 5,233 | 28,630 |

|  | $m_d$ , $M_w$ | $M_L$ , $M_w$ | $m_b$ , $M_w$ | $M_s$ , $M_w$ | M , $M_w$ |
|---|---|---|---|---|---|
| ≠0 | 2,764 | 4,598 | 3,747 | 3,093 | 312 |

## 3. Catalogue homogenization and completeness

### 3.1 Refining the dataset

The dataset is refined in detail for regression analyses to obtain the empirical relations between the magnitudes. In the first step, the catalogue is declustered using Reasenberg's (1985) second-order moment approximation because removing aftershocks is necessary to determine reliable magnitude completeness. For aftershock analysing in space, a subsequent event is searching in an area with a radius of 20 times of the circular source dimension of the preceding event considering ±4 km hypocentre uncertainties (Kanamori and Anderson, 1975; Reasenberg, 1985). The maximum interaction period for the

next event in a sequence is 10 days to build a temporal extension for a cluster. After declustering, the earthquakes occurred after 1980 are selected because the national station networks and data analyses procedure become much more reliable in Turkey. In the third step, completeness (Mc) for each magnitude is determined and it is found that Mc is about ~2.8 for $m_d$ and $M_L$, ~4.0 for $m_b$ and $M_s$. The earthquakes with averaged magnitudes are smaller than the Mc thresholds are excluded in the regressions. In the last step, a cut-off value is applied for high differences between magnitude pairs. There are, naturally,

differences among the reported magnitudes for an earthquake. Occasionally, the difference between the magnitude pairs may be as large as 2 or more magnitude units. After obtaining the distribution of the differences for each pair, the data points that are out of the 95% confidence interval (±2σ) are removed using the Interquartile Range (IQR) method (Galton, 1869; MacAlister, 1879), which is one of the robust methods for outliers and can be successfully applied to seismological data (i.e. Tan et al., 2010, 2014; Tan, 2013). The cut-off values are determined as 0.85 ($m_d$-$M_w$), 0.72 ($M_L$-$M_w$), 0.68 ($m_b$-$M_w$), 1.17

($M_s$-$M_w$), and 1.10 (M-$M_w$). These cut-offs overcome the scattering of the pairs. After refining the magnitude pairs in the four steps, the number of data used in regression is 2100, 3098, 1691, 881 and 228 for $m_d$-$M_w$, $M_L$-$M_w$, $m_b$-$M_w$, $M_s$-$M_w$ and M-$M_w$, respectively (Fig. 4).




### 3.2 Regression Analyses

The relationships of the refined magnitude pairs are estimated using the general orthogonal regression (GOR). The method is
better estimator than the least-square (LS) approximation when both $x$ and $y$ variables have errors of non-negligible size
(Castellaro et al., 2006). The slope ($a$) and intercept ($b$) value of the GOR line in the form of $y = a \cdot x + b$ is given by

$$a = \frac{S_Y^2 - \eta S_X^2 + \sqrt{(S_Y^2 - \eta S_X^2) + 4\eta S_{XY}^2}}{2S_{XY}} \tag{1}$$

$$b = \bar{Y} - a\bar{X} \tag{2}$$


where $S_X^2$, $S_Y^2$ and $S_{XY}^2$ are the covariance of $X$ (independent variable), $Y$ (dependent variable) and between $X$ and $Y$,
respectively (i.e. Castellaro et al., 2006; Das et al., 2014). $\bar{X}$ and $\bar{Y}$ are the average values of the variables. $\eta$ is the error
variance ratio of the variables ($\sigma\varepsilon_X$, $\sigma\varepsilon_Y$) and defined as $\eta = (\sigma\varepsilon_X / \sigma\varepsilon_Y)^2$. When the standard errors of the variables are not
known, $\eta$ is arbitrary set to a value. In practice, $\eta = 1$ (squared Euclidean distance) gives good results (Castellaro et al.,
2006; Das et al., 2014). In this study, $\eta$ is tested for the values from 0.5 to 2.0 to seek a better fit. The $R^2$ values do not
increase when $\eta$ is assigned different than 1.0 and a significant improvement is not observed in the regressions. Besides, the
real errors of the magnitudes are not known; $\eta = 1$ is used. The squared Euclidean distance gives better results for all
magnitude scales. The 95% confidence intervals of the best-fit lines are determined with the bootstrap method (Efron, 1979).
Total 1,000 new regressions are calculated using 50% of the total number of data of each relation. The bootstrap samples are
randomly selected using the Mersenne Twister random number generator (Matsumoto and Nishimura, 1998), and the
random numbers are unique in each test to prevent multiple selections of any datum. After obtaining a large set of the
constants $a$ and $b$ of the linear fits, the outliers are removed with the IQR method. Then, the standard deviation ($\sigma$) of the
normally distributed dataset is calculated.

The GOR results are given in Table 3 and Fig. 4. Because the number of magnitude pairs is high for each relation, the data is
shown with coloured density contours in 0.1 magnitude-unit grids. It is clear that all relations are linear, and minimum misfit
regression lines are in good agreement with the data distribution. The number of pairs is generally dense between the
magnitude of 3.0 and 5.0 and decrease for larger magnitudes. In general, the slopes of the regression lines are close to 1, and
the intercept values are negative except for $M_s$ magnitude. The relation between $m_d$ and $M_w$ indicates, both scale is equal at
$m_d = 4$ and the difference increases up to 0.4 magnitude unit at larger values. $M_L$ values are dense between 3 and 5, and the
linear fitting line extends close to the y=x line. The difference between local and moment magnitudes is about 0.25 at $M_L =$
7.0. The conversion of $m_d$-$M_w$ is similar to that of $M_L$-$M_w$. The largest difference between two different magnitude scales is
observed for surface and moment magnitudes. $M_s$ is always smaller than $M_w$ and the difference is about 0.6 at $M_s = 4.0$. Both
scales are equal at $M_s = 7.5$. The magnitude M (the real type is not known) is mostly reported in the past. There are 95 events




only with M ≥ 5.0 before 1964 in the study area. Therefore, an M-$M_w$ conversion is necessary for seismic hazard analyses using long-term seismicity data. There are few magnitude pairs (N = 228) and they distribute sparsely between 3 and 7 with high standard deviation (Fig. 4).

**Table 3.** Equivalent moment magnitude ($M_w^*$) relations for different magnitude scales.

| Relation | a ±2σ | b ±2σ | Number of Data | Magnitude Range | $R^2$ |
|---|---|---|---|---|---|
| $M_w^* = a \cdot m_d + b$ | +1.125 ±0.025 | -0.507 ±0.102 | 2,100 | 2.8 – 7.3 | 0.80 |
| $M_w^* = a \cdot M_L + b$ | +1.053 ±0.015 | -0.105 ±0.059 | 3,098 | 2.8 – 7.2 | 0.87 |
| $M_w^* = a \cdot m_b + b$ | +1.042 ±0.015 | -0.118 ±0.072 | 1,691 | 4.0 – 7.0 | 0.86 |
| $M_w^* = a \cdot M_s + b$ | +0.838 ±0.035 | +1.213 ±0.164 | 881 | 4.0 – 7.7 | 0.80 |
| $M_w^* = a \cdot M + b$ | +1.057 ±0.108 | -0.199 ±0.551 | 228 | 3.4 – 6.9 | 0.69 |


### 3.3 Homogenization

The GOR results are implemented to all events in the study area. First $M_w$ is searched and assigned as $M_w^*$ if found. For the events without $M_w$, the first averaged magnitude with non-zero value is chosen according to the priority of saturation order
in Table 1. The chosen magnitude is named $M_x$ and used to calculate the equivalent moment magnitude ($M_w^*$) with the relevant equation. After applying homogenization equations to all earthquakes, the catalogue is presented with a total of 45 parameters described in Appendix B. The catalogue has three sections: Event Origin Section, Magnitude Section and Comments. There are 23 parameters in the origin section. The time, coordinates and depths with their uncertainties are given. If one of these parameters is fixed, it is marked with the "f" flag. The magnitude section contains the average with standard
deviation and median of the six magnitude scales. The selected $M_x$ value, its source magnitude scale and calculated equivalent moment magnitude ($M_w^*$) is presented. The ISC event ID number and the epicentre region are given in the comment section as the reference.

In the homogenised catalogue, 55% of the origin parameters are flagged as "PRIME" by the ISC. The ISC and EMSC
(CSEM) origin parameters are generally reported with the prime flag (~90-98%). On the other hand, approximately half of the reported parameters (~40-45%) by the national institutes in Turkey (KOERI, AFAD/DAD) and Greece (ATH) have the flag. The catalogue contains the origin information from the national sources (Fig. 5a) in a high number of percentages. The distribution of the magnitude scales for the equivalent magnitude calculation is given in Fig. 5b. The vast majority $M_w^*$ are obtained from $M_L$ and $m_d$; the contribution of the other magnitude scales is small.


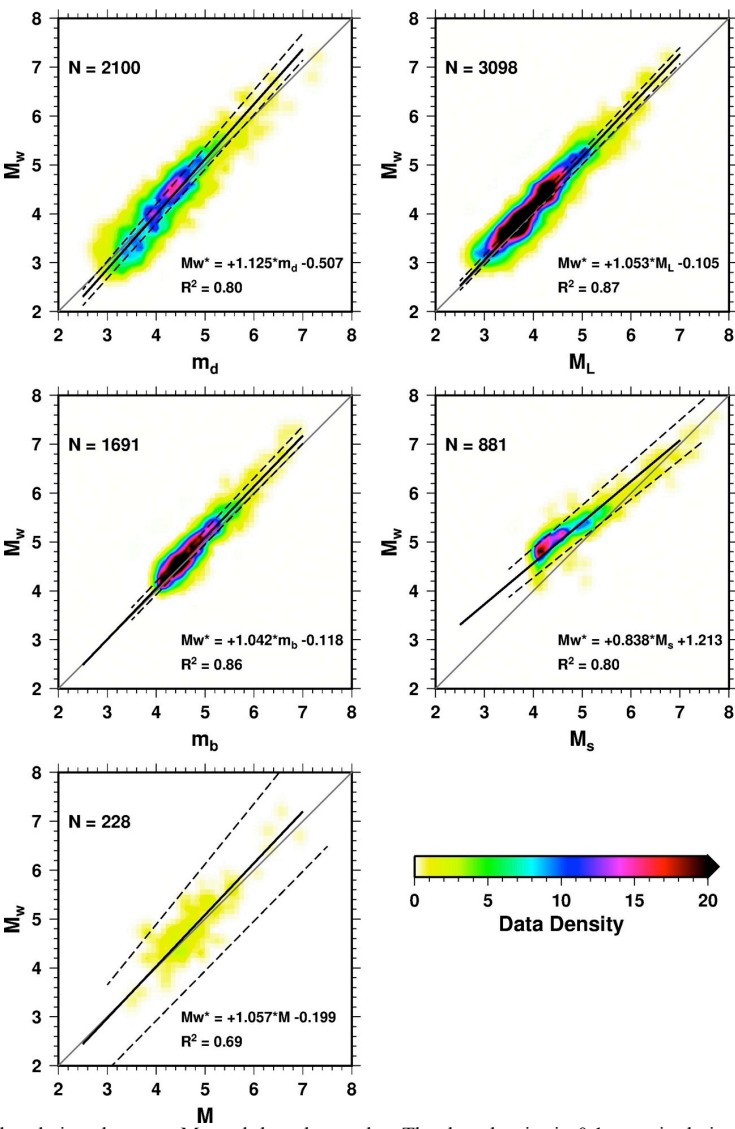

**Figure 5.** Magnitude relations between $M_w$ and the other scales. The data density in 0.1 magnitude intervals is shown with coloured counters. N is the total number of magnitude pairs. The solid line is the best linear fit of the orthogonal regression, whereas the dashed lines shows the 95% confidence interval after bootstrapping. Gray line indicates y=x relation.




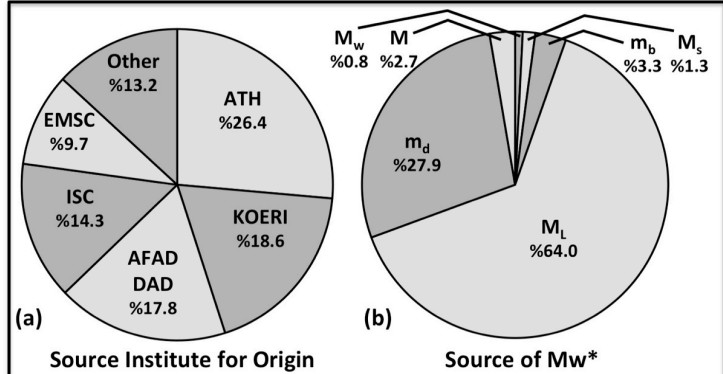

**Figure 6.** Distribution of the parameters in the homogenised catalogue. (a) Source institutes for the origin parameters (b) Magnitude scales used for $M_w$* calculation.


### 3.4 Completeness of the Catalogue

One of the important parameters of an earthquake catalogue is the magnitude of completeness (Mc). Mc is a threshold magnitude and indicates that the earthquakes with magnitudes greater than Mc are recorded in a study area. It is determined using Gutenberg-Richter's (1954) cumulative frequency-magnitude law (GR). The GR relation is simple but powerful and

formulated as $log (N) = a - b \cdot m$, where $N$ is the cumulative number of events with magnitudes equal to or greater than $m$. The other useful parameter derived from this equation is the *b-value* (slope). The *b-value* is around 1 for the tectonically active areas.

The instrumental period (since 1964) observation for the region shows a linear relation with $b = 0.91$ between the cumulative

number of earthquakes and equivalent moment magnitude, $M_w$*, (Fig. 6). If the dataset is extended to cover pre-instrumental period (1900-1964), the linearity for the magnitudes between 5 and 7 due to the magnitude calculation uncertainties of the earthquakes in that time span. The Mc, the lowest intercept point of the linear fit with slope $b$, is 2.9 for the over all catalogue.





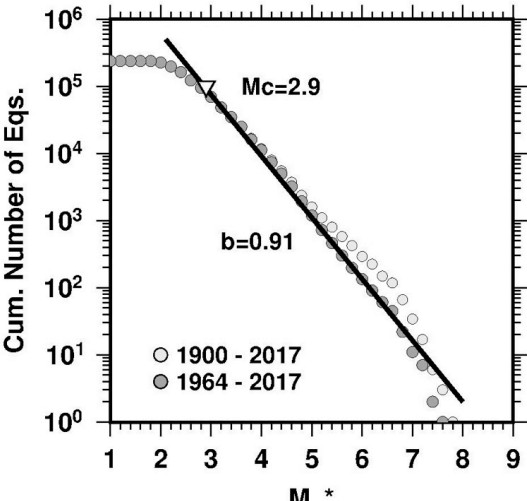

**Figure 7.** Gutenberg-Righter frequency-magnitude distribution for the homogenised catalogue. White circles represents the earthquakes occurred between 1900 and 2017, grey circles represent the earthquakes later than 1964 with the b-value of 0.91. The pre-instrumental period earthquakes cause a bend for the events with ~5-7 $M_w^*$. The catalogue completeness (Mc) is 2.9.

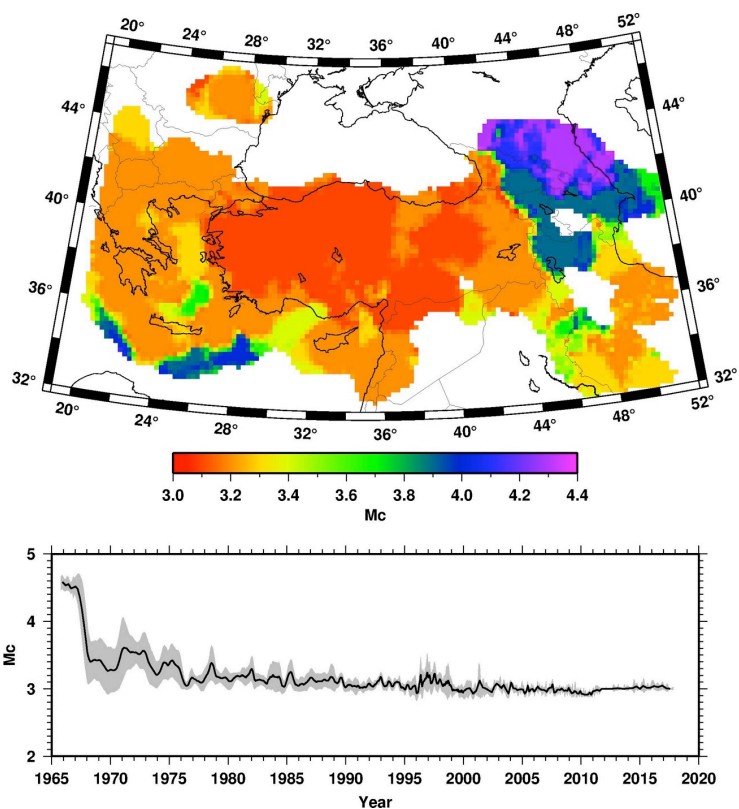

**Figure 8.** Top: Mc spatial distribution map for the events after 1964. There is no data in white areas. Bottom: Temporal Mc variation as a function of year. Grey area is the ±σ interval estimated by bootstrapping.


The maximum curvature method (Wyss et al., 1999; Wiemer, 2001) is applied to investigate the spatial and temporal change of Mc for the instrumental period. Equal horizontal sampling in latitude and longitude is not used to prevent artificial elongation because the length of 1° of longitude is ~94 and ~76 km in south and north of the study area, respectively. I use 20 km grid spacing and at least 100 events larger than the completeness magnitude ($M_w^* > 2.9$) in 100 km radius for the

spatial distribution of Mc. On the other hand, the temporal variation is estimated using a window with 500 events and a step of 25 events. These sampling parameters are sufficient to avoid erroneous statistical results for the *b-value* and Mc due to under-sampling and non-homogenous subsets (Amorese et al., 2010; Kagan, 1999, 2002, 2010; Kamer and Heimer, 2013; Shi and Bolt, 1982). The contour map given in Fig. 8 shows that the homogenised catalogue is complete down to Mw* 3.0-3.2 in Turkey and 3.2-3.3 in Greece. Mc increases dramatically up to 4.0-4.5 in the Caucasus and its abrupt transition follows



the eastern border of Turkey. The regional difference reflects the observation power of the seismological networks. The white areas have very low seismic activity (see Fig. 2) and there is not sufficient data to ensure the criteria. The variation of Mc along years indicates that the standard deviation band is narrow after 1990 and Mc is stable at about 3.0 later than 2000 because the number seismological stations increase after the devastating earthquakes ($M_w > 7$) in 1999.

**4 Discussions**

Generating an earthquake catalogue is the main issue for seismologist. An institution that operates a costly seismological network provides the main parametric information of an event from raw waveform observations. The parametric catalogues are released in paper prints before the internet and are online anymore. Although accessing catalogues is very easy via the internet, it is difficult to obtain all available data due to some limitations of the data providers' web pages. The problems of online datasets, such as absence or limited observation for the past years, a limited number of parameters, lack of parameter

uncertainties, listing limitations, useless formats in web pages etc., make difficult to use the earthquake data for a large range of users. However, most of the researchers pay only attention to the homogenised magnitudes and the number of events. Unfortunately, the importance of a large number of parameters and their uncertainties in a catalogue are missed, and the given datasets less useful for the studies other than seismic hazard analyses.

The earthquake information for Turkey comes from two national networks operated by the KOERI and AFAD. Both institutes have a large number of stations around Turkey and report recent events online. The date, time, depth and magnitudes without uncertainties of events are given by the search engines of both institutions. While the KOERI lists only 50k events in a single search with a downloadable text file, the AFAD search result is given with maximum 100 events at each window and can be downloaded in the comma-separated CSV format. Another online catalogue with the same

parameters is provided by the EMSC. The searched events can be downloaded in the CSV format with the limitations of 5k lines. Among the three institutions, only the KOERI provides all available magnitude scales for an event. Additionally, the EMSC does not provide the type of magnitude scale for an event. On the contrary, the ISC provides all available parameters for an event determined not only by itself but also by the other institutions as mentioned in the previous chapter. The magnitudes in the ISC event list are given in separate lines, so it is not easy to use without knowledge of the comprehensive

bulletin format and programming. The online bulletin search of the ISC has also output limitation with 60k events.

Besides the online catalogues, some catalogue compilations based on homogenization of magnitudes for Turkey and its vicinity are published. Leptokaropoulos et al. (2013) statistically analyse the earthquakes in detail occurred in Western Turkey between (1964-2010) and constructs a catalogue with an equivalent moment magnitude. They obtain conversion

equations for different magnitude scales reported by different institutions. The catalogue contains 9875 events with only parameters of date, time, coordinates and focal depth. Kadirioğlu et al. (2018) present a homogenised catalogue for Turkey





containing ~6573 events between 1900 and 2012. They use the same dataset and conversion equations in their previous study (Kadirioğlu et al., 2014; Kadirioğlu and Kartal, 2016). Their final catalogue is declustered and contains events only $M_w^*$ > 4.0 (not observed $M_w$ as given in the catalogue, notation mistyping). On the other hand, Kadirioğlu et al. (2014, 2018)

mention that a 10 km of focal depth is assigned to the events without reported depth or shallower than 1 km in the final catalogue. This is an arbitrary and unrecoverable parameter assignment and may generate artificial errors in future studies using this catalogue, especially in seismic hazard analyses.

Burton et al. (2004) generate a homogenised catalogue that contains both reported and equivalent magnitudes for

earthquakes in Greece and Western Turkey using the previous conversion equations. There are ~5200 events without Mc analysis. The catalogue by Bayliss and Burton (2007) contains ~3680 homogenised events in Bulgaria and the surrounding Balkan region. It is complete down to 4.0 $M_w$. More recently, Makropoulos et al. (2012) present homogenised event list for calculated $M_s^*$ and $M_w^*$ ~7350 (not observed $M_s$, $M_w$ as given in the catalogue, notation mistyping) in the excel format for Greece and western Turkey.


The common structure of the previous catalogues mentioned above and others has limited earthquake parameters such as date, location, depth and $M_w^*$. Especially, the observed magnitudes and error/uncertainty values are not included. The source institute of the parameters is also missing. Therefore, it is impossible to trace back to the origin of the parameters, and the equivalent moment magnitude ($M_w^*$) cannot be recalculated using newly determined conversion equations. On the other

hand, a truncated final earthquake list using a magnitude threshold is not useful for the researchers who not familiar details of earthquake catalogues and want to analyse or map whole instrumental period seismic activity in a region. The homogenised catalogue overcomes the common deficiency of the previous earthquake catalogues for Turkey and surroundings.

## 5 Conclusions

Turkey and the surrounding area is one of the most seismically active regions on the earth. Therefore, improved earthquake catalogue studies are necessary. A new, extended and homogenised earthquake catalogue is compiled in this study. The main aim is to present an earthquake database in an easily manageable ASCII format for a broad range of researchers. The study is based on the latest ISC Bulletin that its rebuilding process was finished in 2020. All parameters of the earthquakes during the period from 1900 to 2017 in an extended region from the Balkans to the Caucasus are analysed. The origin parameters and

magnitude data in the IASPEI Seismic Format are systematically parsed with a Fortran algorithm.

Approximately 700k events in the study area bounded by 32° - 47° N and 20° - 52° E are compiled (Fig. 3). The equivalent moment magnitude ($M_w^*$), which is the mandatory parameter for the seismic hazard studies, is calculated for all events. For

this purposes, new conversion equations for $m_d$, $M_L$, $m_b$, $M_s$ and M are determined using the well-refined magnitude pairs
using in the general orthogonal regression method that is useful when the two variables have different uncertainties. According to the values of $M_w^*$, the overall catalogue is complete down to Mc = 2.9. The spatial completeness variation indicates Mc = ~3.0-3.2 in Turkey and Greece, and as high as 4.5 in the Caucasus. The catalogue is not declustered or truncated using a threshold magnitude to be useful for geophysicist, geologist and geodesist. The $M_w^*$ values can be easily recalculated and the catalogue can be declustered using different parameters by seismologist and earthquake engineers for
seismic hazard studies. The final dataset contains not only $M_w^*$ as in the previous studies but also the average with standard deviation and median of the observed magnitudes. The ISC event ID-number and geographic region of each event are also given to trace an event in the bulletin. Total of 45 parameters is presented.




**Appendix A**

The ISC contributor institutes mentioned in this study is given below. The ISS and GUTE catalogues are used for pre-instrumental period events.

| Code | Institute |
|------|-----------|
| ISC | International Seismological Centre |
| ISS | International Seismological Summary [for 1900 - 1964] |
| GUTE | Gutenberg and Richter (1954) [for 1900 - 1952] |
| CSEM | European-Mediterranean Seismological Centre - EMSC (France) |
| ISK | B.U. Kandilli Observatory and Earthquake Research Institute (Turkey) |
| DDA | General Directorate of Disaster Affair (Turkey), until Sep.2017 |
| AFAD | Disaster and Emergency Management Presidency (Turkey), since Oct. 2017 |
| ATH | National Observatory of Athens (Greece) |
| THE | Dept. of Geophysics, Aristotle University of Thessaloniki (Greece) |
| MOS | Geophysical Survey of Russian Academy of Sciences (Russia) |
| TEH | Tehran University (Iran) |
| TAB | Tabriz Seismological Observatory (Iran) |
| TIF | Institute of Earth Sciences/ National Seismic Monitoring Center (Georgia) |
| SOF | National Institute of Geophysics, Geology and Geography (Bulgaria) |
| BUC | National Institute for Earth Physics (Romania) |
| SIGU | Subbotin Institute of Geophysics, National Academy of Sciences (Ukraine) |
| IPER | Institute of Physics of the Earth, Academy of Sciences, Moscow (Russia) |
| NSSP | National Survey of Seismic Protection (Armenia) |
| AZER | Republican Seismic Survey Center of Azerbaijan National Academy of Sciences (Azerbaijan) |
| NSSC | National Syrian Seismological Center (Syria) |
| THR | International Institute of Earthquake Engineering and Seismology (Iran) |






**Appendix B**

The first and second lines of the homogenised catalogue are the parameter names and column numbers, respectively. The earthquake parameters are given below.

| | Column | Parameter | | Column | Parameter |
|---|---|---|---|---|---|
| **Event Origin Section** | 1 | Year | **Magnitude Section** | 24 | M (average) |
| | 2 | Mount | | 25 | Std.Dev. of M |
| | 3 | Day | | 26 | M (median) |
| | 4 | Hour | | 27 | $m_d$ (average) |
| | 5 | Minute | | 28 | Std.Dev. of $m_d$ |
| | 6 | Second | | 29 | $m_d$ (median) |
| | 7 | Time Fix Flag | | 30 | $M_L$ (average) |
| | 8 | RMS (s) | | 31 | Std.Dev. of $M_L$ |
| | 9 | Latitude (°) | | 32 | $M_L$ (median) |
| | 10 | Longitude (°) | | 33 | $m_b$ (average) |
| | 11 | Location Fix Flag | | 34 | Std.Dev. of $m_b$ |
| | 12 | Semi-major Axis of 90% ellipse (km) | | 35 | $m_b$ (median) |
| | 13 | Semi-minor axis of 90% ellipse (km) | | 36 | $M_s$ (average) |
| | 14 | Depth (km) | | 37 | Std.Dev. of $M_s$ |
| | 15 | Depth Fix Flag | | 38 | $M_s$ (median) |
| | 16 | Depth Error (km) | | 39 | $M_w$ (average) |
| | 17 | Number of Stations | | 40 | Std.Dev. of $M_w$ |
| | 18 | Azimuthal Gap (°) | | 41 | $M_w$ (median) |
| | 19 | Closest Station Distance (km) | | 42 | $M_x$ |
| | 20 | Furthest Station Distance (km) | | 43 | Source magnitude for $M_x$ |
| | 21 | Event Type | | 44 | $M_w$* |
| | 22 | Institute | **Comments** | 45 | # (null) |
| | 23 | Prime Flag | | 46 | ISC information (event ID and region) |

Fixing Flags:    n: Not fixed (free), f: Fixed

Prime Flags:    n: Not prime location, p: Prime location

Event Types:    de: Damaging earthquake, fe: Felt earthquake,
ke: Known earthquake, se: Suspected earthquake,
uk: Unknown

Unreported numerical parameters in the ISC Bulletin are given as "0.00".

Uncalculated standard deviations are given as "-1.00".

Unknown or blank character fields are filled with "-".





**Data availability**

The catalogue is available as the electronic material of this article.

**Acknowledgments**

I would like to thank Dr. Özlem KARAGÖZ TAN for her support and encouragement for this study. All maps and graphs
are plotted using the Generic Mapping Tools (GMT) by Wessel and Smith (1991).




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
