# Peer review of "A Homogeneous Earthquake Catalogue for Turkey"

_Natural Hazards and Earth System Sciences, 2020_

## Referee Comment (RC1) · Anonymous Referee #1 · 21 Nov 2020

The manuscript provides an earthquake catalogue. According to the title the earthquake catalogue refers to Turkey and Surrounding Region for the instrumental period 1900-2017, which could be of interest to geoscientists. But, the assessed region is too large, including many, event not neighboring to Turkey countries. It is strongly suggested to provide the earthquake catalogue of Turkey (and close surroundings) and not of other neighboring or countries with different seismotectonic regime, such as Albania, Romania, Greece, Syria or Iraq.

Of course, given that the area must be restricted all numbers and percentages mentioned in the manuscript must be updated.

A description of the seismicity map taking into account the seismotectonics is missing. Which are the most seismically active regions and why?

[Figure]

Another question is if the catalogue is really compiled in order to be used in seismic hazard studies, as stated by the author several times. Are the different magnitude scales and the Institutes that calculated the magnitude needed for such studies or by civil engineers? The answer is rather negative. So, the author should analyze the advantages of the proposed catalogue.

Another major issue is that the use of the English language is problematic. The proper use of English language is required. The author should pay attention and be sure to avoid obvious mistakes. Certain points have been corrected but a person with experienced knowledge of the English language should check and correct the manuscript.

Detailed comments, corrections and additions are included, mainly as sticky motes, in the pdf file: nhess-2020-368_reviewer_1

Some of the main comments (also included in the .pdf file) are: 1. Title and region: The title of the paper is not consistent with the selected region. The selected region (32° - 47° N, 20° - 52° E) is too large and not represented by the term "Turkey and Surrounding Region" of the title. It includes all the Balkan countries (e.g. Albania, Bulgaria, Greece, Serbia etc), Caucasus and Arabian countries, a totally inhomogeneous area. This is not "Turkey and Surrounding Region". I strongly suggest to restrict the study area to what the title says, i.e. to the following region: 35° - 44° N, 25° - 46° E.

2. Lines 39-40: "but it cannot be proved this type of man-made faults" What do you mean? Inappropriate use of the English language. Rephrase and explain what you mean.

3. Line 42: "At this point, essential of a homogenised catalogue with a common magnitude arises." What do you mean? Inappropriate use of the English language. Rephrase and explain what you mean. Many other similar cases have been marked at the pdf file.

4. Lines 88-89: "and location procedure that is recently used by the ISC is implemented

to all data." It should be "a location procedure". The author should briefly describe the location procedure.

5. Lines 104-107: "In turn, ISK (Kandilli Observatory and Earthquake Research Institute, KOERI) and DDA (General Directorate of Disaster Affair until September 2017; Disaster and Emergency Management Presidency - AFAD after October 2017), which are the national seismological networks in Turkey, are selected." This is not rational for the huge area selected. Why use the ISK or DDA solution for an event that occurred in Bulgaria or Greece? This selection would be valid if the catalogue was restricted to Turkey, as proposed.

6. Line 116: "On the other hand, we have no evidence for that an institute calculates true magnitude for an earthquake." There are several sentences like this one in the manuscript (e.g. the interested area). All these should be rephrased. The proper use of English language is required. The author must pay attention in order to avoid such obvious mistakes.

7. Line 123: "These limits cover an event that occurred 350 km away from Turkish borders". This is not true. The distances from Turkish borders reach or even exceed 500 km and this has to be changed.

8. Lines 125-126: "The study area also covers the Balkans, Black Sea, Caucasus, Syria, northern Iraq and northwest of Iran. The final catalogue contains 697,139 events occurred in the period from 1900 to the end of 2017." There is no reason to include such a huge and inhomogeneous area. It must be restricted (e.g. to 35° - 44° N, 25° - 46° E). So, all these numbers will change.

9. Lines 137-138: "However, the rate runs up to 6% only in 2010 and 2011 because ∼5000 events without a magnitude are reported by the TIF (Georgia) for the Caucasus earthquakes." Why do you need earthquakes from Georgia and Caucasus? These problems will disappear by changing the study area.

10. Lines: 139-140: "The earthquakes with no magnitude assigned are also included in the catalogue to be useful in future studies." I strongly disagree. The author claims that the proposed catalogue will serve for seismic hazard studies. It is obvious that earthquakes with no magnitude are totally useless for such studies. Earthquakes with no magnitude assigned must be removed.

11. Lines 160-162: "After declustering, the earthquakes occurred after 1980 are selected because the national station networks and data analyses procedure become much more reliable in Turkey." Again, this is not rational for the presented catalogue. It will be correct to do this, only if the catalogue is restricted to Turkey. Why should 1980 be correct for Georgia, Albania or Cyprus? The region must be restricted to Turkey (e.g. to 35° - 44° N, 25° - 46° E).

12. Lines 273-275: "The contour map given in Fig. 8 shows that the homogenised catalogue is complete down to Mw* 3.0- 3.2 in Turkey and 3.2-3.3 in Greece. Mc increases dramatically up to 4.0-4.5 in the Caucasus and its abrupt transition follows the eastern border of Turkey" All the values referring to Greece and Caucasus will be removed if the catalogue is restricted in Turkey, as stated in the title.

13. Lines 287-288: "Unfortunately, the importance of a large number of parameters and their uncertainties in a catalogue are missed, and the given datasets less useful for the studies other than seismic hazard analyses." Apart, from the incorrect use of the English language in this sentence as well, which is a major disadvantage of the submitted manuscript, it has been stated several times that the main scope of the compilation of the proposed catalogue is to use it in seismic hazard studies. Now the contrary is implied, i.e. that other catalogues with less parameters are (which I believe is indeed the case) sufficient for seismic hazard studies. What is the case according to the author? Please clarify.

Please also note the supplement to this comment:
https://nhess.copernicus.org/preprints/nhess-2020-368/nhess-2020-368-RC1-

supplement.pdf

---

## Referee Comment (RC2) · Anonymous Referee #2 · 3 Dec 2020

**Comments**

on the manuscript *"A Homogeneous Earthquake Catalogue for Turkey and Surrounding Region"* by Onur Tan, submitted for publication to *"Natural Hazards and Earth System Sciences"*.

The manuscript is presenting a new earthquake catalog expanding over the period 1900-2017 and covering a very wide region bounded by the coordinates 32° - 47° N, 20° - 52° E.

The compilation of the catalog is based on records of already published earthquake bulletins of international seismological data providers as well as of regional national agencies of Turkey and surrounding countries. There is a special treatment by the author regarding the magnitudes issue in an effort to offer reliable values expressed in the moment magnitude scale. For this reason, five new converting relations are proposed, correlating magnitudes expressed in widely used scales with the moment magnitude scale.

Finally, the magnitude of completeness of the catalog is defined, as well as its variation in space and with time.

Accurate catalogs, with reliable focal parameters (epicenters, focal depths), homogenized regarding the magnitude, are valuable tools, especially for studies regarding seismic hazard assessment. Therefore, the topic of the manuscript is of interest to the readers of the journal. However, in my opinion, there are serious handicaps, which I describe in the following sectors.

In general:

It is not clear which is the procedure followed by the author to adopt the epicenter coordinates and the focal depths for each event of the catalog. Why for earthquakes occurred far away from Turkey the solutions of Turkish seismological centers are considered as more reliable than solutions offered by regional centers operating close to the epicentral area?

Furthermore, there is an extensive description regarding the magnitude homogenization procedure. However, this procedure is not quite clear. Is the finally adopted magnitude coming after a single magnitude conversion following the hierarchy described in the manuscript? Is it a mean value of all available converted magnitudes? Is it a weighted mean?

In addition, there are problems in the quality control of the catalog. Figures 7 and 8 are contradicting each other as in the first one the cut-off magnitude (completeness magnitude) is Mc=2.9 (it is not clear if it corresponds to the period 1964-2017 or 1900-2017) while in the second one, and before ~1995, the Mc is clearly greater than 3.0.

Finally, the on-line part of the catalog is not representative at all. There are not enough cases of recent earthquakes with more than one available magnitudes in order to test the effectiveness of the process followed by the author.

In details:

1) Although English is not my mother tongue, I would say that English throughout the manuscript is quite poor. Bad English made it difficult (and in some cases

impossible) for me to understand certain parts of the manuscript. I recommend the author to check and correct the manuscript in order to make it more understandable to the reviewers and/or to the readers.

2) The region under study is so wide that is far away from been characterized as *"Turkey and surrounding region"*.

3) In this wide region shallow as well as intermediate depth earthquakes occur. It is well known that their records differ significantly from each other, meaning that there is no way the same converting relations to be applicable for both. There is no mention in the manuscript of any particular procedure followed for intermediate depth events.

4) Line 32: What do you mean *"then they are averaged"*? There are several magnitude estimations reported for each event and expressed in different magnitude scales. How these values have been *"averaged"*?

5) Is there any special treatment for events reported as "explosions" or *"mining activities"* or, in general, for artificial events?

6) Line 105: For such a wide area, the process of final selection of focal parameters for adoption is rather delicate and, in any case, is not sufficiently explained in the text. For example, why solutions from Turkish seismological centers should be preferred for earthquakes occurred in distant regions such as Adriatic, Romania etc. instead of solutions of Italian or Romanian institutes?

7) Line 107: *"The other institutes are used for the local events around Turkey"*. This is contradicting with the previous reference.

8) Line 113 – Figure 2: I am confused. In the text, you mention, *"if there are two or more values for each type, average with standard deviation and median are calculated"*. What do you mean? If there are more than one magnitude values reported in the same scale, what you have done? Have you calculated their mean value? If this is the case then, how do you know how these magnitude values correlate to each other?

9) Line 132: It looks that ISC bulletins were used as the source of Mw values. However, ISC does not estimate moment magnitudes, instead, it includes in its bulletins moment magnitudes from other available sources, such as GCMT (former HRVD), NEIC etc. Have you checked their consistency to each other? There are also reports of seismic moment values in reliable catalogs (e.g. Pacheco and Sykes, 1992; Engdahl and Villasenor, 2002; etc.). Have you used them to enrich the available moment magnitudes in your catalog?

10) I strongly disagree with including in the catalog earthquakes with no magnitudes. Usually such earthquakes are not strong enough to give reliable recordings that are necessary for a robust estimation of focus and/or magnitude. In such a case, their focal parameters could be questionable, contaminating the final product.

11) Figure 4: Searching the ISC data-base for the period 1900-2017 and for the region that you have used I found 22,970 mb values reported by ISC and a total of 33,607 reported by ISC & NEIC. The respective numbers of Ms values were 4,557 & 12,716. Even though these numbers do not agree with the respective ones in the histograms of figure 4, it is more likely that you have also used magnitudes other than ISC. Have you checked their compatibility to each other (i.e. mbISC/mbNEIC and MsISC/MsNEIC) before considering them as a priori equivalent?

12) Line 162: What is the reason to check each magnitude scale's completeness in a catalog? The completeness check has a meaning if it is performed in a homogenized (with respect to magnitudes) earthquake catalog in an effort to reveal its quality characteristics.

13) Line 163: What do you mean *"averaged magnitudes"*? How can there be averaged magnitudes for each scale (!) and for each earthquake? It is not comprehensive what exactly is that you have done. Please, clarify.

14) Line 201: This difference is expected, since ML starts underestimating for magnitudes over ~6.0 and undergoes saturation for values over ~6.5 (e.g. Heaton et al., 1986). It has also been shown that Ms exhibits rather bilinear behavior becoming equivalent to Mw for Ms$\geq$6.0 (e.g. Heaton eat al., 1986; Scordilis, 2006). Such a behavior is also visible in the graph of figure 5. You should take it into account.

15) Line 214: What do you mean by *"priority saturation order"*? Which was the procedure applied when there were more than one converted magnitude values available? Have you adopted the converted Mw* value following the hierarchy of table 1? Have you used a mean value of all converted magnitudes? A weighted mean value? You must be clear about that.

16) Lines 228-229: Fig 5a and fig 5b must be renamed to Fig 6a and Fig 6b, respectively.

17) Line 250: What do you mean by the term *"pre-instrumental period (1900-1964)"*? The term is completely inappropriate. There were installed seismographs during this period in the study region. The same expression is also met in the caption of figure 7.

18) In Figure 8b it is obvious that the value Mc=2.9 for the magnitude of completeness does not hold for the whole period. I would say that it could be considered Mc=3.0 since 1995 or Mc=3.1 since 1978 and, maybe, Mc=3.4-3.5 since ~1968. So two maps should replace the map of figure 8: one for the period 1968-1978 and the second for 1978-2017. Relative adjustments are also needed for figure 7.

19) I believe that the sample of 500 events with 480 events overlapping (moving step of 20 events) forms a very strong filter, which "hides" temporal changes of Mc values (Fig 8b).

20) Line 277: The change in detectability of networks after the 1999 Izmit earthquake is not visible, probably due to the strong filtering that has been applied in sampling.

21) In my opinion, the first paragraph of *"Discussion"* is not needed at all. I suggest you delete it.

22) Line 325: *"On the other hand, a truncated final earthquake list using a magnitude threshold is not useful for the researchers who not familiar details of earthquake catalogues and want to analyse or map whole instrumental period seismic activity in a region".* I disagree. Researchers less familiar with data could be misled by using earthquake catalogs with non-complete data. In my opinion, completeness of data must be considered as a prerequisite for a published catalog. However, incomplete data could be included in the catalog, provided they do not have zero magnitudes (equivalent to Mw).

23) Appendix A: It looks that well-known published catalogs, global and regional, have not been considered (e.g. Papazachos and Papazachou, 1997, 2003; Pacheco and Sykes, 1992; Karnik, 1996; Engdahl and Villaseñor, 2002 etc.). They are not even mentioned in the manuscript.

24) Line 411: Correct reference Galton… 1896 to Galton… 1869.

25) Appendix B: In column 2 replace "Mount" with "Month"

---

## Referee Comment (RC3) · Anonymous Referee #3 · 7 Dec 2020

1.Earthquake catalog is an essential tool for seismology and seismology engineer. For seismic hazard analysis, in general, we convert the different magnitude scales to Mw (Harvard, CMT). How do you calculate the Mw* ?  What is the relation between Mw (Harvard, CMT) and Mw* ?

2. After you convert the different magnitudes scale to Mw*, how many events are the magnitudes equal to or great than 6.0 ?

3. As described as 2., how many events are the crust earthquakes ? mid-depth earthquakes ?

4. The Mc is closed related with seismic stations dense, how many seismic stations are installed within the Turkey country ?

---

## Author Comment (AC1) · 16 Feb 2021

Response to the Referee #1

for

"A Homogeneous Earthquake Catalogue for Turkey "

by Onur Tan

**General**

First, I want to thank all referees for their vulnerable comments. I revised the database and manuscript (MS) according to their comments.

The main revisions:

- The title was changed: "A Homogeneous Earthquake Catalogue for Turkey"
- The catalogue area was reduced according to the common comments:   34°-44° N   24°-46° E

[Figure]

Previous area                                   Revised area

- The events in the period of Jan-Oct 2018 were included because ISC updated the database.
- Mw* = 0.0 events were removed from the database.
- The database was reanalysed.
- All numerical outputs, tables, and figures were updated

**Response to Referee #1**

**> > All comments of Referee #1 on the supplementary PDF file (commented manuscript) are considered in detail.**

*The manuscript provides an earthquake catalogue. According to the title the earth- quake catalogue refers to Turkey and Surrounding Region for the instrumental period 1900-2017, which could be of interest to geoscientists. But, the assessed region is too large, including many, event not neighboring to Turkey countries. It is strongly suggested to provide the earthquake catalogue of Turkey (and close surroundings) and not of other neighboring or countries with different seismotectonic regime, such as Albania, Romania, Greece, Syria or Iraq.*

*Of course, given that the area must be restricted all numbers and percentages mentioned in the manuscript must be updated.*

The catalogue area was restricted, and all outputs were updated.

*A description of the seismicity map taking into account the seismotectonics is missing. Which are the most seismically active regions and why?*

I added a short information about seismicity into the Introduction section as follow:

The western Anatolia is the most seismically active part of Turkey. Both N-S extension in Aegean and the westward motion of Anatolian Plate along the NAFZ cause a dense deformation with small to moderate

earthquakes in western Turkey. The North and East Anatolian Fault zones are also the primary seismic sources that generate destructive earthquakes (Mw ≥ 6).

*Another question is if the catalogue is really compiled in order to be used in seismic hazard studies, as stated by the author several times. Are the different magnitude scales and the Institutes that calculated the magnitude needed for such studies or by civil engineers? The answer is rather negative. So, the author should analyze the advantages of the proposed catalogue.*

The catalogue area is restricted, and all events with zero magnitudes are excluded.

The advantage of the catalogue is also mentioned in the Conclusion part.

*Another major issue is that the use of the English language is problematic. The proper use of English language is required. The author should pay attention and be sure to avoid obvious mistakes. Certain points have been corrected but a person with experienced knowledge of the English language should check and correct the manuscript.*

The MS was checked for grammatical errors. The mistakes were corrected.

*Detailed comments, corrections and additions are included, mainly as sticky motes, in the pdf file: nhess-2020-368_reviewer_1*

**> > All comments on the supplementary PDF file (commented manuscript) are considered in detail.**

*Some of the main comments (also included in the .pdf file) are:*

*1. Title and region: The title of the paper is not consistent with the selected region. The selected region (32° - 47° N, 20° - 52° E) is too large and not represented by the term "Turkey and Surrounding Region" of the title. It includes all the Balkan countries (e.g. Albania, Bulgaria, Greece, Serbia etc), Caucasus and Arabian countries, a totally inhomogeneous area. This is not "Turkey and Surrounding Region". I strongly suggest to restrict the study area to what the title says, i.e. to the following region: 35° - 44° N, 25° - 46° E.*

As mentioned in the General section of this document, the study area was restricted (34°-44°N 24°-46°E).

*2. Lines 39-40: "but it cannot be proved this type of man-made faults" What do you mean? Inappropriate use of the English language. Rephrase and explain what you mean.*

This sentence was removed because it was not suitable for the context.

*3. Line 42: "At this point, essential of a homogenised catalogue with a common magnitude arises." What do you mean? Inappropriate use of the English language. Rephrase and explain what you mean. Many other similar cases have been marked at the pdf file.*

The sentences were rephrased as follows:

*One common magnitude scale should be used to standardise analyses in the studies based on the parametric data such as hazard mitigation. Therefore, a homogenized catalogue with a unified magnitude scale becomes essential. In the last two decades, the studies on unifying earthquake magnitudes and generating improved catalogues are carried out for different regions on the Earth (i.e. Grünthal et al., 2009; Chang et al., 2016; Manchuel et al., 2018; Rovida et al., 2020).*

*4. Lines 88-89: "and location procedure that is recently used by the ISC is implemented to all data." It should be "a location procedure". The author should briefly describe the location procedure.*

The ISC location procedure is not applied in this study. Therefore the detail of the ISC process is not mentioned in the manuscript. The sentence is rewritten, and the reference of the location procedure is cited:

*The ISC finished rebuilding the entire database in 2020 by utilizing a new location algorithm (Bondár and Storchak, 2011) with the ak135 seismic velocity model (Kennett et al., 1995). Furthermore, previously unavailable hypocentre and station phase readings from the permanent and temporary networks are added to the rebuild bulletin (ISC, 2020; Storchac et al., 2017). Therefore, the latest and revised international dataset is used in this study.*

*5. Lines 104-107: "In turn, ISK (Kandilli Observatory and Earthquake Research Institute, KOERI) and DDA (General Directorate of Disaster Affair until September 2017; Disaster and Emergency Management Presidency - AFAD after October 2017), which are the national seismological networks in Turkey, are selected." This is not rational for the huge area selected. Why use the ISK or DDA solution for an event that occurred in Bulgaria or Greece? This selection would be valid if the catalogue was restricted to Turkey, as proposed.*

The Turkish seismology agencies (ISK and DDA) do not locate the events in far away from Turkey because the neighbouring countries are out of the networks. Rarely, moderate events in the neighbouring countries are reported by them. Out of the international agencies, e.g. the events in Greece and Bulgaria are reported by Obs. of Athens and Sofia National Institute of Geophysics, respectively. The selection algorithm used in this study (flowchart in Fig. 2) is checking the location (see the map below). If an event far away from the Turkish border is reported by ISK or DDA, the hypocentre parameters of both agencies are omitted and data of the local agencies is selected.

[Figure]

The events located by KOERI (blue) and DDA/AFAD (green) in the homoginesed catalogue.

*6. Line 116: "On the other hand, we have no evidence for that an institute calculates true magnitude for an earthquake." There are several sentences like this one in the manuscript (e.g. the interested area). All these should be rephrased. The proper use of English language is required. The author must pay attention in order to avoid such obvious mistakes.*

Thank you very much. I checked and corrected this type of mistakes.

*7. Line 123: "These limits cover an event that occurred 350 km away from Turkish borders". This is not true. The distances from Turkish borders reach or even exceed 500 km and this has to be changed.*

*8. Lines 125-126: "The study area also covers the Balkans, Black Sea, Caucasus, Syria, northern Iraq and northwest of Iran. The final catalogue contains 697,139 events occurred in the period from 1900 to the*

*end of 2017." There is no reason to include such a huge and inhomogeneous area. It must be restricted (e.g. to 35° - 44° N, 25° - 46° E). So, all these numbers will change.*

*9. Lines 137-138: "However, the rate runs up to 6% only in 2010 and 2011 because ~5000 events without a magnitude are reported by the TIF (Georgia) for the Caucasus earthquakes." Why do you need earthquakes from Georgia and Caucasus? These problems will disappear by changing the study area.*

*10. Lines: 139-140: "The earthquakes with no magnitude assigned are also included in the catalogue to be useful in future studies." I strongly disagree. The author claims that the proposed catalogue will serve for seismic hazard studies. It is obvious that earthquakes with no magnitude are totally useless for such studies. Earthquakes with no magnitude assigned must be removed.*

*11. Lines 160-162: "After declustering, the earthquakes occurred after 1980 are selected because the national station networks and data analyses procedure become much more reliable in Turkey." Again, this is not rational for the presented catalogue. It will be correct to do this, only if the catalogue is restricted to Turkey. Why should 1980 be correct for Georgia, Albania or Cyprus? The region must be restricted to Turkey (e.g. to35 -44N,25-46E).*

*12. Lines 273-275: "The contour map given in Fig. 8 shows that the homogenised catalogue is complete down to Mw\* 3.0- 3.2 in Turkey and 3.2-3.3 in Greece. Mc increases dramatically up to 4.0-4.5 in the Caucasus and its abrupt transition follows the eastern border of Turkey" All the values referring to Greece and Caucasus will be removed if the catalogue is restricted in Turkey, as stated in the title.*

Because the area is restricted, the comments in #7, 8, 9, 11, and 12 are disappeared.

The events with no magnitude are excluded from the catalogue as commented in #10. I agree with the referee.

According to the similar comments from the other referees, I did not use an Mc cut-off for the spatial distribution calculation in the revised version. After adding new events in Jan-Oct 2018, I re-calculated the b-value and Mc for the period of 1964-2018. Fig 7 and 8 were updated.

[Figure]

[Figure]

Fig 7 – previous          Fig 7 - updated

*13. Lines 287-288: "Unfortunately, the importance of a large number of parameters and their uncertainties in a catalogue are missed, and the given datasets less useful for the studies other than seismic hazard analyses." Apart, from the incorrect use of the English language in this sentence as well, which is a major disadvantage of the submitted manuscript, it has been stated several times that the main scope of the compilation of the proposed catalogue is to use it in seismic hazard studies. Now the contrary is implied, i.e. that other catalogues with less parameters are (which I believe is indeed the case) sufficient for seismic hazard studies. What is the case according to the author? Please clarify.*

I am sorry for the discrepancy in the sentences. According to my experience in the SSHAC Level-2 for the Sinop Nuclear Power Plant (Turkey), the uncertainties of all available parameters must be included in the homogenised catalogue. I used the same steps given in Fig. 2 to prepare the catalogue for the power plant.

This part was rewritten as follows:

*Unfortunately, the importance of providing more parameters and their uncertainties in the previous catalogues are missed. For example, the SSG-9 (item #3.27i) safety document of the International Atomic Agency for nuclear power plant requires the uncertainties of all earthquake parameters.  Therefore, the previously given datasets are less useful, especially for seismic hazard analyses.*

---

## Author Comment (AC2) · 16 Feb 2021

**Response to the Referee #2**
**for**
**"A Homogeneous Earthquake Catalogue for Turkey "**
**by Onur Tan**

**General**

First, I want to thank to all referees for their vulnerable comments. I revised the database and manuscript (MS) according to their comments.

The main revisions:

- The title was changed: "A Homogeneous Earthquake Catalogue for Turkey"
- The catalogue area was reduced according to the common comments:   34°-44° N   24°-46° E

[Figure]

Previous area                              Revised area

- The events in the period of Jan-Oct 2018 were included because ISC updated the database.
- Mw* = 0.0 events were removed from the database.
- The database was reanalysed.
- All numerical outputs, tables, and figures were updated

**Response to Referee #2**

*on the manuscript "A Homogeneous Earthquake Catalogue for Turkey and Surrounding Region" by Onur Tan, submitted for publication to "Natural Hazards and Earth System Sciences".*

*The manuscript is presenting a new earthquake catalog expanding over the period 1900-2017 and covering a very wide region bounded by the coordinates 32° - 47° N, 20° - 52° E.*

*The compilation of the catalog is based on records of already published earthquake bulletins of international seismological data providers as well as of regional national agencies of Turkey and surrounding countries. There is a special treatment by the author regarding the magnitudes issue in an effort to offer reliable values expressed in the moment magnitude scale. For this reason, five new converting relations are proposed, correlating magnitudes expressed in widely used scales with the moment magnitude scale.*

*Finally, the magnitude of completeness of the catalog is defined, as well as its variation in space and with time.*

*Accurate catalogs, with reliable focal parameters (epicenters, focal depths), homogenized regarding the magnitude, are valuable tools, especially for studies regarding seismic hazard assessment. Therefore, the*

*topic of the manuscript is of interest to the readers of the journal. However, in my opinion, there are serious handicaps, which I describe in the following sectors.*

Thank you for your thoughts.

***In general:***

*It is not clear which is the procedure followed by the author to adopt the epicenter coordinates and the focal depths for each event of the catalog. Why for earthquakes occurred far away from Turkey the solutions of Turkish seismological centers are considered as more reliable than solutions offered by regional centers operating close to the epicentral area?*

A location procedure was not applied in this study. I used the revised ISC database in 2020. The sentences at the end of the first paragraph of Section 2 were rewritten and the location procedure reference was cited.

The Turkish seismology agencies (ISK and DDA) do not locate the events in far away from Turkey because the neighbouring countries are out of the networks. Rarely, moderate events in the neighbouring countries are reported by them. Out of the international agencies, e.g. the events in Greece and Bulgaria are reported by Obs. of Athens and Sofia National Institute of Geophysics, respectively. The selection algorithm used in this study (flowchart in Fig. 2) is checking the location (see the map below). If an event far away from the Turkish border is reported by ISK or DDA, the hypocentre parameters of both agencies are omitted and data of the local agencies is selected.

[Figure]

The events located by KOERI (blue) and DDA/AFAD (green) in the homogenised catalogue.

On the other hand, as mentioned in the General section above, the study area was restricted (34°-44° N   24°-46° E).

*Furthermore, there is an extensive description regarding the magnitude homogenization procedure. However, this procedure is not quite clear. Is the finally adopted magnitude coming after a single magnitude conversion following the hierarchy described in the manuscript? Is it a mean value of all available converted magnitudes? Is it a weighted mean?*

The averaged values are for the reported magnitudes. For example, if there are six reported ML values for an event, their arithmetic mean is calculated without weighting.

The sentences are re-written for clarity as below. The flowchart in Fig. 2 is also updated.

*After determining the event origin parameters in the selected area, the magnitude data sub-block is analysed by the magnitude parser. The reported values of different magnitude scales given in Table 1 are collected. If there are two or more values for a magnitude scale, the arithmetic mean and median of all reported values are calculated. Selecting a magnitude value from a particular institute such as KOERI, Harvard, and EMSC is not preferred to overcome the problems such as unreported magnitude, the effect of network distribution, and calculation errors.*

*In addition, there are problems in the quality control of the catalog. Figures 7 and 8 are contradicting each other as in the first one the cut-off magnitude (completeness magnitude) is Mc=2.9 (it is not clear if it corresponds to the period 1964-2017 or 1900- 2017) while in the second one, and before ~1995, the Mc is clearly greater than 3.0.*

The Mc in Fig 7 was calculated for the events in the large area from1964 to 2017. In the first version of the MS, I used the Mc=2.9 value as a cut-off value in the spatial distribution of the Mc. Therefore, minimum Mc in the map is about 3.

Thank you for this valuable comment.

In the revised version, I did not use a cut-off for the spatial distribution calculation.

After adding new events in Jan-Oct 2018, I re-calculated the b-value and Mc for the period of 1964-2018. Fig 7 and 8 were updated.

[Figure]

[Figure]

Fig 7 – previous                    Fig 7 - updated

[Figure]

[Figure]

Fig 8 – previous                    Fig 8 - updated

*Finally, the on-line part of the catalog is not representative at all. There are not enough cases of recent earthquakes with more than one available magnitudes in order to test the effectiveness of the process followed by the author.*

Because of the open system of the journal, I do not prefer the upload the full version of the catalogue.

*In details:*

*1) Although English is not my mother tongue, I would say that English throughout the manuscript is quite poor. Bad English made it difficult (and in some cases impossible) for me to understand certain parts of the manuscript. I recommend the author to check and correct the manuscript in order to make it more understandable to the reviewers and/or to the readers.*

The MS was checked for grammatical errors. The mistakes were corrected.

*2) The region under study is so wide that is far away from been characterized as "Turkey and surrounding region".*

The catalogue area was restricted and all outputs were updated.

*3) In this wide region shallow as well as intermediate depth earthquakes occur. It is well known that their records differ significantly from each other, meaning that there is no way the same converting relations to be applicable for both. There is no mention in the manuscript of any particular procedure followed for intermediate depth events.*

In the revised catalogue, the intermediate depths are mostly excluded from the catalogue by narrowing the study area. The high percentage of the events is in Turkey. Therefore, additional conversion equations for different depth intervals are not defined.

*4) Line 32: What do you mean "then they are averaged"? There are several magnitude estimations reported for each event and expressed in different magnitude scales. How these values have been "averaged"?*

This part was re-written as given above.

*5) Is there any special treatment for events reported as "explosions" or "mining activities" or, in general, for artificial events?*

No. The magnitudes of artificial seismic events are not larger than $M_L$ 1.5-2.0 in Turkey. KOERI and DAD/AFAD identify explosions and do not include earthquake catalogue since 2005. However, the blasts in the earlier years are questionable. For example, KOERI reports blasts in a different catalogue. In this study, the events with an explosion flag in the ISC bulletin are not selected.

If there is a real blast that is not identified in the ISC Bulletin, it is eliminated before the regression because of the Mc-threshold.

I added the sentence below to Section 3.1.

"Using a threshold helps eliminate a possible blast (M < 2.0-2.5) before the regression."

*6) Line 105: For such a wide area, the process of final selection of focal parameters for adoption is rather delicate and, in any case, is not sufficiently explained in the text. For example, why solutions from Turkish seismological centers should be preferred for earthquakes occurred in distant regions such as Adriatic, Romania etc. instead of solutions of Italian or Romanian institutes?*

As I mentioned above the parameters of an event in the neighbouring country are obtained from the international (i.e. ISC, EMSC) or the local agencies (i.e Obs. of Athens).

*7) Line 107: "The other institutes are used for the lfocal events around Turkey". This is contradicting with the previous reference.*

The sentence was re-written.

*8) Line 113 – Figure 2: I am confused. In the text, you mention, "if there are two or more values for each type, average with standard deviation and median are calculated". What do you mean? If there are more than one magnitude values reported in the same scale, what you have done? Have you calculated their mean value? If this is the case then, how do you know how these magnitude values correlate to each other?*

For example, if there are 4 $M_L$ values for an event, the average and median of them calculated.   The sentence is re-written as follow:

> *After determining the event origin parameters in the selected area, the magnitude data sub-block is analysed by the magnitude parser. The reported values of different magnitude scales given in Table 1 are collected. If there are two or more values for a magnitude scale, the arithmetic mean and median of all reported values are calculated. Selecting a magnitude value from a particular institute such as KOERI, Harvard, and EMSC is not preferred to overcome the problems such as unreported magnitude, the effect of network distribution, and calculation errors.*

*9) Line 132: It looks that ISC bulletins were used as the source of Mw values. However, ISC does not estimate moment magnitudes, instead, it includes in its bulletins moment magnitudes from other available sources, such as GCMT (former HRVD), NEIC etc. Have you checked their consistency to each other? There are also reports of seismic moment values in reliable catalogs (e.g. Pacheco and Sykes, 1992; Engdahl and Villasenor, 2002; etc.). Have you used them to enrich the available moment magnitudes in your catalog?*

Yes, it is true that the ISC does not determine Mw for an event. The institutions' Mw estimations and their consistencies are not the scopes of this study. Whether two (or more) Mw (or other scales) values are consistency or not for an event, they are reported and are in the international databases. If the values are close, their standard deviations are small in the homogenised catalogue (column #40). The std.dev. show the consistency.

I do not prefer to use printed papers in this study. For a standard database and format, the ISC Bulletin is preferred.

*10) I strongly disagree with including in the catalog earthquakes with no magnitudes. Usually such earthquakes are not strong enough to give reliable recordings that are necessary for a robust estimation of focus and/or magnitude. In such a case, their focal parameters could be questionable, contaminating the final product.*

I agree with the referee.

The events with no magnitudes are excluded from the catalogue.

*11) Figure 4: Searching the ISC data-base for the period 1900-2017 and for the region that you have used I found 22,970 mb values reported by ISC and a total of 33,607 reported by ISC & NEIC. The respective numbers of Ms values were 4,557 & 12,716. Even though these numbers do not agree with the respective ones in the histograms of figure 4, it is more likely that you have also used magnitudes other*

*than ISC. Have you checked their compatibility to each other (i.e. mbISC/mbNEIC and MsISC/MsNEIC) before considering them as a priori equivalent?*

Of course, I used the all reported, e.g. mb values for an event. Not only ISC & NEIC but also EMSC and national observatories in the region. As I mentioned in the manuscript, I averaged all values for each scale. Here is an example:

```
Event   1768311 Turkey
   Date       Time      Err   RMS Latitude Longitude  Smaj  Smin  Az Depth    Err Ndef Nsta Gap  mdist  Mdist Qual    Author      OrigID
2000/11/07 21:13:58.49  0.70 1.461  39.4287   26.2702 4.007 3.863  32  10.0f      77   69  24   0.18  23.57 m i se ISC          4370817
(#PRIME)

 Magnitude
 Err Nsta Author      OrigID
Mb    3.6          NAO       3610951
mb    4.0       3 NEIC       4036975
MD    3.9          ISK       4036975
ML    3.4          THE       4036975
ML    4.0          ATH       4036975
MD    3.8          ISK       3041816
MD    4.0      12 ATH        4015830
ML    4.0          ATH       4015830
ML    3.7          THE       3860907
```

The average mb value comes from the reports of NOA and NEIC. The ML is from THE and ATH. There is no way to analyse multiple reports from an institution. Therefore, averaging is a good way to assign one mb and ML value to the event.

*12) Line 162: What is the reason to check each magnitude scale's completeness in a catalog? The completeness check has a meaning if it is performed in a homogenized (with respect to magnitudes) earthquake catalog in an effort to reveal its quality characteristics.*

I used Mc of each magnitude scale to obtain more reliable data set for conversion equations. It is also a good tool to exclude possible blasts.

*13) Line163:What do you mean "averaged magnitudes"? How can there be averaged magnitudes for each scale (!) and for each earthquake? It is not comprehensive what exactly is that you have done. Please, clarify.*

The average magnitude calculation is mentioned in Section 2.

For example, all $M_L$ values and all Mw values of an event are averaged because there is no way to construct a reliable relation for individual values. To obtain a single $M_L$ and Mw pair for each event, the average value is the best way according to my opinion.

*14) Line 201: This difference is expected, since ML starts underestimating for magnitudes over ~6.0 and undergoes saturation for values over ~6.5 (e.g. Heaton et al., 1986). It has also been shown that Ms exhibits rather bilinear behavior becoming equivalent to Mw for Ms>6.0 (e.g. Heaton eat al., 1986; Scordilis, 2006). Such a behavior is also visible in the graph of figure 5. You should take it into account.*

I try to give a single equation for each conversion for simplicity. The recent dataset in Fig5 show a linear relation between MS and Mw. On the other hand, bilinear behaviour is in the uncertainty interval.

*15) Line 214: What do you mean by "priority saturation order"? Which was the procedure applied when there were more than one converted magnitude values available? Have you adopted the converted Mw\* value following the hierarchy of table 1? Have you used a mean value of all converted magnitudes? A weighted mean value? You must be clear about that.*

I need only a one conversion equation for Mw* calculation. Otherwise, different Mw* values can be calculated for an event. This is an ambiguity for the users. The best tool is the saturation of magnitude scales.   The sentence below was added to Section 3.3  for clarity.

*For example, if an event has only average Ms and ML values, Ms is selected for Mw* calculation.*

Some one also calculate Mw* using other scales because the catalogue has all values. The catalogue gives a flexible usage.

*16)  Lines 228-229: Fig 5a and fig 5b must be renamed to Fig 6a and Fig 6b, respectively.*

It is corrected.

*17)  Line250:What do you mean by the term "pre-instrumental period (1900-1964)"? The term is completely inappropriate. There were installed seismographs during this period in the study region. The same expression is also met in the caption of figure 7.*

The terms are corrected.

"pre-instrumental period (1900-1964)" is changed to "*the period from 1900 to 1964*"

"*The modern instrumental period  …*"  is used for the period since 1964.

*18)  In Figure8b it is obvious that the value Mc=2.9 for the magnitude of completeness does not hold for the whole period. I would say that it could be considered Mc=3.0 since 1995 or Mc=3.1 since 1978 and, maybe, Mc=3.4-3.5 since ~1968. So two maps should replace the map of figure 8: one for the period 1968-1978 and the second for 1978-2017. Relative adjustments are also needed for figure 7.*

Because the study area is narrowed, all graphs are changed.

I think that the maps for different periods give similar information with Fig8b.

Mc distribution in the study area between 1964 and 1978 is given below. Because there are few located events in the area, Mc calculation at each grid is not accurate.

[Figure]

Earthquakes (1964-1978)                                             G-R plot

[Figure]

Mc map for the years 1964-1978. There is no enough data for b-value and Mc analyses.

*19) I believe that the sample of 500 events with 480 events overlapping (moving step of 20 events) forms a very strong filter, which "hides" temporal changes of Mc values (Fig 8b).*
*20) Line277:The change in detect ability of networks after the 1999 Izmit earthquake is not visible, probably due to the strong filtering that has been applied in sampling.*

Thank you for these comments (#19-20).

I changed the parameters. I used 200 events with 40-event-step. It is much more appropriate.  Now it is clearly seen that the network improvement since 2007.

[Figure]

Fig 8b – previous                                Fig 8b - updated

*21) In my opinion, the first paragraph of "Discussion" is not needed at all. I suggest you delete it.*

The first paragraph is introduction information for the later discussion. I think that it is better to hold the paragraph.

*22) Line 325: "On the other hand, a truncated final earthquake list using a magnitude threshold is not useful for the researchers who not familiar details of earthquake catalogues and want to analyse or map whole instrumental period seismic activity in a region". I disagree. Researchers less familiar with data could be misled by using earthquake catalogs with non-complete data. In my opinion, completeness of*

*data must be considered as a prerequisite for a published catalog. However, incomplete data could be included in the catalog, provided they do not have zero magnitudes (equivalent to Mw).*

The zero-magnitude events are excluded from the catalogue.

The catalogue is not only for seismic hazard studies. For example, a geologist wants to plot a seismicity map for a region in Turkey. He/she may want to see small events in the region. He/she can truncate the data to plot bigger events.

*23) AppendixA: It looks that well-known published catalogs, global and regional, have not been considered (e.g. Papazachos and Papazachou, 1997, 2003; Pacheco and Sykes, 1992; Karnik, 1996; Engdahl and Villaseñor, 2002 etc.). They are not even mentioned in the manuscript.*

Only the ISC bulletin is considered for a standard data procedure. The printed event lists are not suitable for the used process in this study.

*24) Line 411: Correct reference Galton... 1896 to Galton... 1869.*

It is corrected.

*25) Appendix B: In column 2 replace "Mount" with "Month"*

It is corrected.

---

## Author Comment (AC3) · 16 Feb 2021

**Response to the Referee #3**
**for**
**"A Homogeneous Earthquake Catalogue for Turkey "**
**by Onur Tan**

**General**

First, I want to thank to all referees for their vulnerable comments. I revised the database and manuscript (MS) according to their comments.

The main revisions:

- The title was changed: "A Homogeneous Earthquake Catalogue for Turkey"
- The catalogue area was reduced according to the common comments: 34°-44° N   24°-46° E

[Figure]

Previous area                                  Revised area

- The events in the period of Jan-Oct 2018 were included because ISC updated the database.
- Mw* = 0.0 events were removed from the database.
- The database was reanalysed.
- All numerical outputs, tables, and figures were updated

**Response to Referee #3**

*1. Earthquake catalog is an essential tool for seismology and seismology engineer. For seismic hazard analysis, in general, we convert the different magnitude scales to Mw (Harvard, CMT). How do you calculate the Mw* ? What is the relation between Mw (Harvard, CMT) and Mw* ?*

The individual Mw estimations of the institutions are not the scope of this study. In my opinion, either Harvard or another institution is not a monopoly for Mw calculations. Therefore, I am against to use the parameters of one institution for magnitude homogenisation.

I prepared the Mw* catalogue with the same strategy for the Sinop Nuclear Power Plant (Turkey) and it was accepted in the SSHAC Level-2.

*2. After you convert the different magnitudes scale to Mw*, how many events are the magnitudes equal to or great than 6.0 ?*

There are 179 events occurred in the newly defined area. I added the number to the caption of Fig 3:

*Figure 3. The earthquakes in the homogenised catalogue (dots). Yellow circles are the events with an equivalent moment magnitude (Mw*) greater than 6.0 (total 179 events).*

*3. As described as 2., how many events are the crust earthquakes? mid-depth earthquakes ?*

The study area was narrowed according to the referees' common comments.

There are only 795 events with h>40 km and Mw*>4) in the new catalogue area. It is very small part of the catalogue.

*4. The Mc is closed related with seismic stations dense, how many seismic stations are installed within the Turkey country?*

The number of active stations in Turkey is about 1240. Kandilli observatory (KOERI) has 116 weak-motion and 113 strong-motion stations. AFAD has 309 weak-motion and 679 strong-motion stations.

I mentioned the total number of the station in Turkey:

> *The earthquake information for Turkey comes from two national networks operated by the KOERI and AFAD. Both institutes have a large number of stations around Turkey (~1240) and report recent events online.*

---

## Author Response (AR3)

**Response to Referee #1**

*The manuscript provides an earthquake catalogue for Turkey for the instrumental period 1900-2018, which is of interest to geoscientists.*

*The author performed most of the suggested changes to the manuscript, the most important of which was the restriction of the study area.*

*However, the area that was suggested for the catalogue was 35° - 44° N, 25° - 46° E. On the contrary, the author chose the area 34°-44° N 24°-46° E which again has to be restricted as already suggested to the following: 35° - 44° N, 25° - 46° E.*

*Of course, given that the area must be restricted again all numbers and percentages mentioned in the manuscript must be also updated.*

*When the author performs this change, the manuscript will be eligible for publication.*

The catalogue area was restricted for the final version, and all outputs were updated. The boundary of the final area is  35° - 44° N, 25° - 46° E.

[Figure]

1st version                          2nd version

Final Version

The full data file of the final catalogue is uploaded.